# Two-Stage Robust Optimization for Prosumers Considering Uncertainties from Sustainable Energy of Wind Power Generation and Load Demand Based on Nested C&CG Algorithm

Qiang Zhou [1], Jianmei Zhang [1], Pengfei Gao [1], Ruixiao Zhang [1], Lijuan Liu [1], Sheng Wang [1], Lin Cheng [2], Wei Wang [2] and Shiyou Yang [2,*]

1    Gansu Key Laboratory of Renewable Energy Integration Operation and Control, State Grid Gansu Electric Power Research Institute, Lanzhou 730070, China; eezhouqiang@163.com (Q.Z.); zhangjm_dky@163.com (J.Z.); gpf910724@163.com (P.G.); changjuihsiao@foxmail.com (R.Z.); 18294452085@163.com (L.L.); 18521733306@163.com (S.W.)
2    College of Electrical Engineering, Zhejiang University, Hangzhou 310027, China; 22210087@zju.edu.cn (L.C.); d77mvp@zju.edu.cn (W.W.)
*    Correspondence: eesyyang@zju.edu.cn; Tel.: +86-0571-87952498

**Abstract:** This paper develops a two-stage robust optimization (TSRO) model for prosumers considering multiple uncertainties from the sustainable energy of wind power generation and load demand and extends the existing nested column-and-constraint generation (C&CG) algorithm to solve the corresponding optimization problem. First, considering the impact of these uncertainties on market trading strategies of prosumers, a box uncertainty set is introduced to characterize the multiple uncertainties; a TSRO model for prosumers considering multiple uncertainties is then constructed. Second, the existing nested C&CG algorithm is extended to solve the corresponding optimization problem of which the second-stage optimization is a bi-level one and the inner level is a non-convex optimization problem containing 0–1 decision variables. Finally, a case study is solved. The optimized final overall operating cost of prosumers under the proposed model is CNY 3201.03; the extended algorithm requires only four iterations to converge to the final solution. If a convergence accuracy of $10^{-6}$ is used, the final solution time of the extended algorithm is only 9.75 s. The case study result shows that prosumers dispatch the ESS to store surplus wind power generated during the nighttime period and release the stored electricity when the wind power generation is insufficient during the daytime period. It can contribute to promoting the local accommodation of renewable energy and improving the efficiency of renewable energy utilization. The market trading strategy and scheduling results of the energy storage system (ESS) are affected by multiple uncertainties. Moreover, the extended nested C&CG algorithm has a high convergence accuracy and a fast convergence speed.

**Keywords:** prosumer; nested column-and-constraint generation algorithm; two-stage robust optimization; multiple uncertainties

## 1. Introduction

To solve the world energy shortage and environmental pollution, there is an increasingly urgent demand on renewable energy or sustainable energy [1]. As a result, the penetration rate of distributed new energy sources on the energy consumption side has been increasing. However, with the increasing penetration level of renewable energy, the uncertainties of renewable energy generation will cause a greater impact on the safe and stable operation of the power grid, and the renewable energy accommodation will also become more prominent. In this regard, energy storage technology plays a significant role in improving the level of renewable energy accommodation and maintaining the safe and stable operation of the power system. There have been numerous studies devoted to the

energy storage system (ESS). In [2], a comprehensive study was reported for determining the battery size in the battery ESS. In [3], the optimal installation of the battery ESS and capacitors was investigated using the optimization objective of maximizing the benefits of the low-voltage distribution system with a high percentage of photovoltaic (PV), and the application of a battery ESS was proved to significantly improve the voltage distribution of the power system. In [4], it was presented that the overvoltage in low-voltage power grids with a high PV penetration can be prevented through the application of an ESS.

With the continuous advancement of distributed renewable energy generation technology, the number of prosumers with "source-load" dual attributes, which can deliver electricity to the grid, is gradually increasing in the distribution network [5]. Prosumers are generally an aggregate of multiple distributed energy sources, such as customer-side distributed sources, the ESS, and electric vehicles [6,7]. Prosumers can independently manage their internal flexibility resources and achieve efficient management and control of load-side resources by virtue of the flexibility and complementarity of their aggregation units [5,8]. Prosumers can participate in electricity market transactions as independent interest subjects [9]. Numerous efforts have been devoted to energy management and energy trading strategies for prosumers. In [10], an energy management strategy for large-scale prosumer groups based on the interactive energy mechanism was proposed. In [11], a market trading strategy and an energy management method among community prosumers were proposed based on the master–slave game model. In [12], a multi-objective optimization model based on time-of-day tariff was proposed to optimize the operating cost and market trading electricity quantity of prosumers through the cooperative scheduling of "source-load-storage". In [13], a bi-auction market mechanism was proposed to optimize both the operation and market trading strategies of prosumers. In [14], a peer-to-peer (P2P) energy trading model using the objective of the total energy cost of all smart households in a microgrid was established, and a near-optimal energy cost optimization algorithm was proposed. In [15], a P2P energy trading mechanism for sharing the ownership of an ESS among multiple users in a residential community was investigated. In [16], an energy trading scheme with the participation of sustainable users, considering both fairness and optimality among prosumers, was introduced. In [17], a low-carbon P2P energy trading model considering clean energy preferences of prosumers was investigated. In [18], a P2P energy trading model on the basis of an urban community microgrid information–physical network system, considering the coordination and complementarity of flexibility resources among prosumers, was presented.

It should be pointed out that in the existing works of [10–18], the impact of the uncertainties of renewable energy generation or sustainable energy on energy management and energy trading strategies of prosumers are not considered. Considering the fact that prosumers are equivalently the active load with a high percentage of renewable energy, it is necessary to fully consider the impact of the uncertainties of renewable energy generation on the overall benefits of prosumers in developing energy management and energy trading strategies. Generally, robust optimization (RO), stochastic optimization (SO), and distributed robust optimization (DRO) are typical approaches to deal with optimization problems under uncertainties. In [19], an optimal scheduling strategy for prosumers considering participation in joint energy market transactions was presented, and an RO method was used to address the uncertainty of PV generations. In [20], a day-ahead energy trading model for prosumers equipped with PV facilities was established, and an adjustable RO method was investigated to address the uncertainty of PV generations. In [21], a two-stage robust optimization (TSRO) model for prosumers considering the uncertainties of renewable energy generation and the electricity market price was developed, and the column-and-constraint generation (C&CG) algorithm was introduced to solve the linearized optimization problem. In [22], a two-stage SO model was proposed for prosumers equipped with PV and battery ESS facilities. In [23], an energy sharing strategy for neighboring PV prosumers based on an energy sharing provider equipped with an ESS was presented, and an SO method was used to deal with the uncertainty in

the PV generation, the electricity price, and the load demand. In [24], a DRO approach was proposed to address the uncertainties of renewable energy generations for prosumers. In [25], a data-driven DRO model for P2P energy trading among prosumers was developed. In addition, there are some other studies that consider multiple uncertainties in their models. In [26,27], scenarios under multiple uncertainties caused by renewable energy and load were modeled using the corresponding probability distribution functions. In [28], multiple uncertainties aroused by renewable energy and load were addressed by the Latin hypercube sampling (LHS) method. In [29], multiple scenarios were generated using the LHS method. In [30], the uncertainties of harmonic source load were addressed under the proposed probability distribution model. However, in the existing approaches, the RO method relies heavily on the calculation scenarios. As the number of scenarios increases, the number of models will increase exponentially, which brings the curse of dimensionality and thus leads to a low solving efficiency. Moreover, a DRO model is a typical non-convex optimization model, which is difficult to transform into a convex problem. With the increase in uncertainty factors, the complexity of the model will also increase. An RO method can obtain the optimal robust strategy under all possible worst-case scenarios without any requirement on the probability distribution information of uncertainties. It can flexibly adjust the robustness and conservatism of the model by changing the robust parameter, and the original problem can be equivalently transformed using the C&CG algorithm, which has a high solving efficiency. In addition, a single-stage RO model is generally too conservative and pessimistic, so it is better to use a TSRO method to deal with the uncertainties in the optimization problems for prosumers.

However, most of these studies on energy management and energy trading optimization are devoted to solving a preconditioned TSRO problem, which is where the second-stage optimization is a convex optimization problem, and none have been directed to cases where the second-stage optimization of the energy management or energy trading optimization problem for prosumers is a bi-level one and the inner level is a non-convex optimization problem containing 0–1 decision variables. The models developed in these studies either do not consider the decision of the ESS in the second-stage optimization or do not consider the case where the ESS of prosumers cannot be charged and discharged simultaneously in the second-stage optimization problem.

Therefore, on the basis of the existing works, this paper considers the impact of multiple uncertainties from the sustainable energy of wind power generation and load demand on market trading strategies and scheduling results of an ESS. It establishes a general TSRO model for prosumers using the optimization objective of minimizing the overall operating cost of prosumers, where the second-stage optimization is a bi-level one and the inner level is a non-convex optimization problem containing 0–1 decision variables. In addition, the model considers the scenarios where the ESS of prosumers cannot be charged and discharged simultaneously in the second-stage optimization problem. Moreover, considering that the optimization in the second stage of the established model is a bi-level one and the inner level is a non-convex optimization problem containing 0–1 decision variables, the existing nested C&CG algorithm [31] is extended and used to solve the proposed model. Finally, the results of the market trading strategy and optimal scheduling results of the ESS considering multiple uncertainties are reported using case studies, and the convergence and effectiveness of the introduced algorithm are verified.

The remainder of this paper is organized as follows: In Section 2, the TSRO model for prosumers using the optimization objective of minimizing the overall operating cost is established. In Section 3, the nested C&CG algorithm is extended to solve the proposed TSRO model. In Section 4, a case study is solved, and the optimization results are presented, compared, and discussed. In Section 5, the conclusions are abstracted, and the limitations of the current study and suggestions for future work are suggested.

## 2. A TSRO Model for Prosumers Considering Multiple Uncertainties

Prosumers are generally an aggregate of customer-side flexibility resources. In this paper, the aggregation units of prosumers include customer-side wind power generation, ESS, and fixed load. Then, a box uncertainty set is introduced to characterize multiple uncertainties [32–34]. Finally, this paper develops a TSRO model considering multiple uncertainties using the optimization objective of minimizing the overall operating cost of prosumers.

### 2.1. Objective Function

For prosumers aggregated by wind power generation, the ESS, and fixed load, the optimization objective of the proposed model considering multiple uncertainties includes specifically the revenue from electricity sales, the cost of electricity purchase, and the cost of the ESS. Therefore, the objective function is formulated as follows:

$$\min_{\mathbf{Y}_{\mathbf{Non}}^{\mathbf{t}}} \left[ \sum_{t=1}^{T} \left( C_t^{buy} - R_t^{sell} \right) + \max_{\mathbf{U}_{\mathbf{res}}, \mathbf{U}_{\mathbf{Load}}} \min_{\mathbf{X}_{\mathbf{Non}}^{\mathbf{t}}} \sum_{t=1}^{T} C_t^{ESS} \right], \tag{1}$$

where $T$ is the dispatch period. $C_t^{buy}$ and $R_t^{sell}$ are the cost of electricity purchase and the revenue from electricity sales at time $t$, respectively. Since the ESS of the prosumers will cause battery loss in charging and discharging, which will affect the battery lifetime [35], the cost of battery loss should be considered in the cost of the ESS, and $C_t^{ESS}$ is the cost of the ESS for power charging and discharging at time $t$. $\mathbf{U}_{\mathbf{res}}$ and $\mathbf{U}_{\mathbf{Load}}$ are both uncertainty sets. $\mathbf{Y}_{\mathbf{Non}}^{\mathbf{t}}$ and $\mathbf{X}_{\mathbf{Non}}^{\mathbf{t}}$ are the vector sums of decision variables in the first and the second stage, $\mathbf{Y}_{\mathbf{Non}}^{\mathbf{t}} = [P_{Pb}^t, P_{Ps}^t, Z_{Pb}^t, Z_{Ps}^t]$ and $\mathbf{X}_{\mathbf{Non}}^{\mathbf{t}} = [P_{Ec}^t, P_{Ed}^t, SOC^t, Z_{Ec}^t, Z_{Ed}^t]$, respectively.

2.1.1. Revenue from Electricity Sales

The revenue from electricity sales is

$$R_t^{sell} = \mu_{Ps}^t P_{Ps}^t, \tag{2}$$

where $\mu_{Ps}^t$ is the price for selling electricity of prosumers at time $t$, and $P_{Ps}^t$ is the electrical energy sold by prosumers at time $t$.

2.1.2. Cost of Electricity Purchase

The cost of electricity purchase is given as

$$C_t^{buy} = \mu_{Pb}^t P_{Pb}^t, \tag{3}$$

where $\mu_{Pb}^t$ is the price for purchasing electricity of prosumers at time $t$, and $P_{Pb}^t$ is the electrical energy purchased by prosumers at time $t$.

2.1.3. Cost of ESS

The cost of ESS is calculated using

$$C_t^{ESS} = c_E \left( P_{Ec}^t + P_{Ed}^t \right), \tag{4}$$

where $c_E$ is the coefficient of the charged and discharged costs of ESS, and $P_{Ec}^t$ and $P_{Ed}^t$ are the charged and discharged power of ESS at time $t$, respectively.

### 2.2. Constraints
2.2.1. Constraints on Power Balance

It is mandatory to ensure the power balance by applying

$$P_{Pb}^t + P_{res}^t + P_{Ed}^t - P_{Ec}^t - P_{Ps}^t - P_{Load}^t = 0, \tag{5}$$

where $P_{res}^t$ is the wind power generation of prosumers at time $t$, and $P_{Load}^t$ is the internal load demand of prosumers at time $t$.

### 2.2.2. Constraints on Purchased/Sold Power

The constraints on purchased/sold power include

$$0 \leq P_{Pb}^t \leq P_{Pb}^{\max} Z_{Pb}^t \tag{6}$$

$$0 \leq P_{Ps}^t \leq P_{Ps}^{\max} Z_{Ps}^t \tag{7}$$

$$Z_{Pb}^t + Z_{Ps}^t \leq 1, \tag{8}$$

where $P_{Pb}^{\max}$ and $P_{Ps}^{\max}$ are the maximum purchased and sold power of prosumers, respectively; $Z_{Pb}^t$ and $Z_{Ps}^t$ are both 0–1 variables, representing whether prosumers purchase and sell electricity at time $t$, respectively.

### 2.2.3. Constraints on the ESS

The constraints on the ESS include constraints on the charged and discharged power of the ESS and constraints on the capacity of the ESS.

(1)  The constraints on charged and discharged power of the ESS are given as

$$0 \leq P_{Ec}^t \leq P_{Ec}^{\max} Z_{Ec}^t \tag{9}$$

$$0 \leq P_{Ed}^t \leq P_{Ed}^{\max} Z_{Ed}^t \tag{10}$$

$$Z_{Ec}^t + Z_{Ed}^t \leq 1, \tag{11}$$

where $P_{Ec}^{\max}$ *and* $P_{Ed}^{\max}$ are the maximum charged and discharged power of ESS, respectively; $Z_{Ec}^t$ and $Z_{Ed}^t$ are both 0–1 variables, representing whether ESS is charged and discharged at time $t$, respectively.

(2)  The constraints on the capacity of the ESS are formulated as

$$SOC^1 - SOC^{init} - P_{Ec}^1 \eta^{Ec} + \frac{P_{Ed}^1}{\eta^{Ed}} = 0 \tag{12}$$

$$SOC^t - SOC^{t-1} - P_{Ec}^t \eta^{Ec} + \frac{P_{Ed}^t}{\eta^{Ed}} = 0 \tag{13}$$

$$SOC^{24} - SOC^{init} = 0 \tag{14}$$

$$SOC^{\min} \leq SOC^t \leq SOC^{\max}, \tag{15}$$

where $SOC^t$ is the quantity of electricity stored in the ESS at time $t$, $SOC^{init}$ is the initial quantity of electricity stored in the ESS, $\eta^{Ec}$ and $\eta^{Ed}$ are the charged and discharged efficiencies of the ESS, respectively, and $SOC^{\max}$ and $SOC^{\min}$ are the maximum and minimum values of the capacity of the ESS, respectively.

### 2.3. Uncertainty Sets

Because of the random and fluctuating characteristics of wind speed, wind power generation, a sustainable energy, has strong uncertainties [36,37]. In addition, due to the influence of social, economic, and environmental issues, the actual load demand of prosumers will exhibit random fluctuations and presents substantial uncertainties [38–40].

Therefore, prosumers need to take into account the influence of multiple uncertainties caused by wind power generation and load demand in their day-ahead electricity market trading decisions.

In this paper, multiple uncertainties are described as box uncertainty sets. First, the forecasted data of wind power generation and load demand are obtained on the basis of their historical data. Then, their respective maximum fluctuation ranges are predicted. Finally, the robust parameter is set to obtain the uncertainty sets, which are represented as follows:

$$
\mathbf{U_{res}} = \left\{ \mathbf{P_{res}} \in \mathbf{R}^{1 \times T} : \sum_{t=1}^{T} \left( Z_{res,Lb}^{t} + Z_{res,Ub}^{t} \right) \leq \Gamma, \right.
$$
$$
\Delta_{res,Lb}^{t} = 0.2 Z_{res,Lb}^{t} P_{res,pre}^{t},
$$
$$
\Delta_{res,Ub}^{t} = 0.2 Z_{res,Ub}^{t} P_{res,pre}^{t},
$$
$$
P_{res,pre}^{t} - \Delta_{res,Lb}^{t} \leq P_{res}^{t} \leq P_{res,pre}^{t} + \Delta_{res,Ub}^{t},
$$
$$
\left. Z_{res,Lb}^{t} + Z_{res,Ub}^{t} \leq 1, \forall t \in T \right\} \tag{16}
$$

$$
\mathbf{U_{Load}} = \left\{ \mathbf{P_{Load}} \in \mathbf{R}^{1 \times T} : \sum_{t=1}^{T} \left( Z_{Load,Lb}^{t} + Z_{Load,Ub}^{t} \right) \leq \Gamma, \right.
$$
$$
\Delta_{Load,Lb}^{t} = 0.2 Z_{Load,Lb}^{t} P_{Load,pre}^{t},
$$
$$
\Delta_{Load,Ub}^{t} = 0.2 Z_{Load,Ub}^{t} P_{Load,pre}^{t},
$$
$$
P_{Load,pre}^{t} - \Delta_{Load,Lb}^{t} \leq P_{Load}^{t} \leq P_{Load,pre}^{t} + \Delta_{Load,Ub}^{t},
$$
$$
\left. Z_{Load,Lb}^{t} + Z_{Load,Ub}^{t} \leq 1, \forall t \in T \right\}, \tag{17}
$$

where $Z_{res,Lb}^{t} / Z_{Load,Lb}^{t}$ and $Z_{res,Ub}^{t} / Z_{Load,Ub}^{t}$ are both 0–1 variables, representing whether the fluctuation range of wind power generation/load demand reaches the minimum and maximum values of the forecast error at time $t$, respectively. $P_{res,pre}^{t}$ and $P_{Load,pre}^{t}$ are both the forecasted data at time $t$. $\Gamma$ is the robust parameter, which is used for conservative adjustment.

### 3. Solution Methodology Based on a Nested C&CG Algorithm

*3.1. Overview of the Proposed Solution Methodology*

Considering that the second-stage optimization of the constructed model is a bi-level one and the inner level optimization is non-convex, containing 0–1 decision variables, the nested C&CG algorithm is extended to solve the proposed optimization model. First, the model is decomposed into a main problem and a subproblem, and they are solved alternately, which is called the outer-level C&CG process. Then, considering that the subproblem is a bi-level one and the inner level is a non-convex optimization problem containing 0–1 decision variables, the subproblem is broken down into a subproblem and a main problem, and they are also solved alternately, which is called the inner-level C&CG process. By solving the model with the introduced algorithm, one can obtain the optimal scheduling results that minimize the operating cost of prosumers under the worst case of multiple uncertainties. The simplified block diagram of the proposed solution methodology is shown in Figure 1.

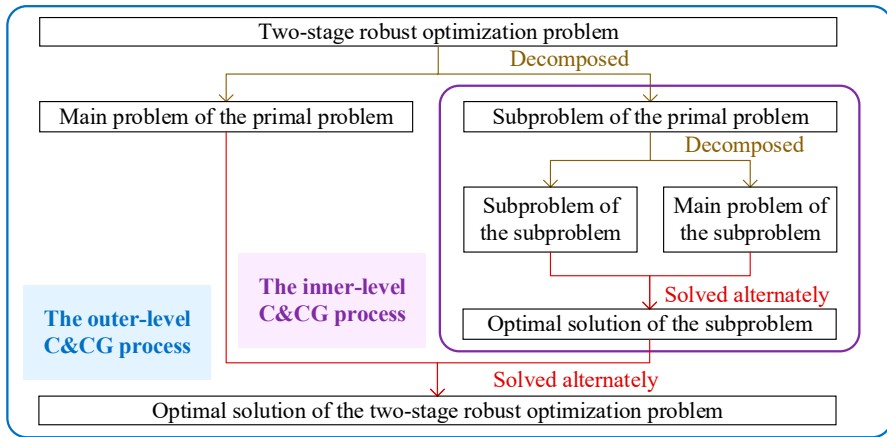

**Figure 1.** The simplified block diagram of the proposed solution methodology.

### 3.2. C&CG Algorithm

The general TSRO model expressed in a compact matrix form is presented as follows:

$$\begin{cases} \min_{\mathbf{y}} \left( \mathbf{cy} + \max_{\mathbf{u} \in \mathbf{U}} \min_{\mathbf{x}} \mathbf{dx} \right) \\ s.t. \; \mathbf{Ay} \le \mathbf{b} \\ \qquad \mathbf{Ey} + \mathbf{Fx} \le \mathbf{h} - \mathbf{Ru}, \end{cases} \tag{18}$$

where **c**, **d**, **A**, **b**, **E**, **F**, **h**, and **R** represent the coefficient matrix. **y** and **x** both represent decision variable matrices, and **x** of the second stage does not contain 0–1 decision variables. **u** represents the uncertain variable matrix, and **U** is the uncertainty set.

In addition, the subproblem of the second stage of the model is essentially a piecewise block convex function about x, as shown in Figure 2.

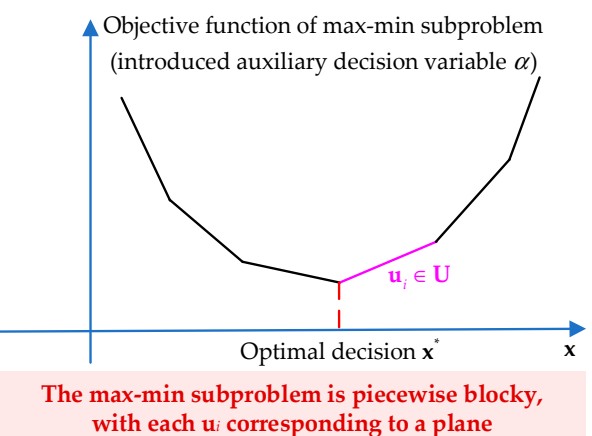

**Figure 2.** The max-min subproblem.

Therefore, the C&CG algorithm is introduced [41]. It breaks down model (18) into a min main problem and a max-min subproblem, then solves them alternately to acquire the optimal solution.

### 3.2.1. Min Main Problem

The min main problem is used for solving the first-stage optimization, and its model is formulated as follows:

$$\begin{cases} \min_{\mathbf{y},\alpha,\mathbf{x}}(\mathbf{cy}+\alpha) \\ s.t. \ \alpha \geq \mathbf{dx}^p, \forall p \leq q \\ \quad \mathbf{Ay} \leq \mathbf{b} \\ \quad \mathbf{Ey}+\mathbf{Fx}^p \leq \mathbf{h}-\mathbf{Ru}_p, \forall p \leq q \end{cases} \tag{19}$$

where $q$ is the number of iterations. $\alpha$ is an introduced auxiliary decision variable. $\mathbf{x}^p$ is the decision variable matrix added to the min main problem at the $p$th iteration. $\mathbf{u}_p$ is the worst-case scenario of uncertain variables acquired by solving the max-min subproblem at the $p$th iteration.

### 3.2.2. Max-Min Subproblem

After the first-stage decision $\mathbf{y}^*$ is obtained in the min main problem, the max-min subproblem is used for solving the second-stage optimization, and its model is given as follows:

$$\begin{cases} \max_{\mathbf{u}\in\mathbf{U}}\min_{\mathbf{x}}\mathbf{dx} \\ s.t. \ \mathbf{Ey}^*+\mathbf{Fx} \leq \mathbf{h}-\mathbf{Ru}. \end{cases} \tag{20}$$

### *3.3. Nested C&CG Algorithm*

The TSRO model for prosumers considering multiple uncertainties constructed in this paper is

$$\begin{cases} \min_{\mathbf{Y}_{\mathbf{Non}}^{\mathbf{t}}}\left[\sum_{t=1}^{T}\left(\mu_{Pb}^t P_{Pb}^t - \mu_{Ps}^t P_{Ps}^t\right) + \max_{\mathbf{U_{res}},\mathbf{U_{Load}}}\min_{\mathbf{X}_{\mathbf{Non}}^{\mathbf{t}}}\sum_{t=1}^{T}c_E\left(P_{Ec}^t+P_{Ed}^t\right)\right] \\ s.t. \ (5)-(17). \end{cases} \tag{21}$$

For convenience with comparison to model (20), all decision variables in model (21) are then converted into the matrix vector form, and the following compact model is obtained:

$$\begin{cases} \min_{\mathbf{y}}\left(\mathbf{cy}+\max_{\mathbf{u}\in\mathbf{U}}\min_{\mathbf{x},\mathbf{z}}\mathbf{dx}\right) \\ s.t. \ \mathbf{Ay} \leq \mathbf{b} \\ \quad \mathbf{Ey}+\mathbf{Fx}+\mathbf{Gz} \leq \mathbf{h}-\mathbf{Ru} \\ \quad \mathbf{Jz} \leq \mathbf{v}, \end{cases} \tag{22}$$

where $\mathbf{c}$, $\mathbf{d}$, $\mathbf{A}$, $\mathbf{b}$, $\mathbf{E}$, $\mathbf{F}$, $\mathbf{G}$, $\mathbf{h}$, $\mathbf{R}$, $\mathbf{J}$, and $\mathbf{v}$ represent the coefficient matrix. $\mathbf{y}$, $\mathbf{x}$, and $\mathbf{z}$ all represent decision variable matrices, and $\mathbf{z}$ is a 0–1 decision variable matrix of the second stage. $\mathbf{u}$ represents the uncertain variable matrix, and $\mathbf{U}$ is the uncertainty set.

Different from the general model (18), the model constructed in (22) of this paper contains 0–1 decision variables in its second-stage decision variable matrix, so the constructed model (22) cannot be directly solved by the C&CG algorithm. Therefore, the nested C&CG algorithm is introduced to solve the constructed model (22).

### 3.3.1. Outer-Level C&CG Procedures

The constructed model (21) is decomposed into a main problem and a subproblem of the original problem, which can be solved alternately to acquire the optimization results.

(1)  Main Problem of the Original Problem

The optimization results of the first stage can be acquired by solving the main problem. At the $l$th iteration of the outer level, the worst-case scenario $\mathbf{u}_l$, $\mathbf{u}_l = [P_{res}^{t,l}, P_{Load}^{t,l}]$, of uncertain variables is given by solving the subproblem at the previous iteration. Then,

decision variables $\mathbf{x}^l$ and $\mathbf{z}^l$ and their corresponding constraints are added, $\mathbf{x}^l = [P_{Ec}^{t,l}, P_{Ed}^{t,l}, SOC^{t,l}]$ and $\mathbf{z}^l = [Z_{Ec}^{t,l}, Z_{Ed}^{t,l}]$. Finally, the optimization results of the first-stage optimization can be acquired. The model of the main problem is presented as follows:

$$
\begin{cases}
\min\limits_{\eta, \mathbf{Y}_{\mathbf{Non}}^{\mathbf{t}}} \left[ \sum\limits_{t=1}^{T} \left( \mu_{Pb}^t P_{Pb}^t - \mu_{Ps}^t P_{Ps}^t \right) + \eta \right] \\
s.t.\ \eta \geq \sum\limits_{t=1}^{T} c_E \left( P_{Ec}^{t,l} + P_{Ed}^{t,l} \right), \forall l \leq k \\
P_{Pb}^t + P_{res}^{t,l} + P_{Ed}^{t,l} - P_{Ec}^{t,l} - P_{Ps}^t - P_{Load}^{t,l} = 0, \forall l \leq k \\
0 \leq P_{Ec}^{t,l} \leq P_{Ec}^{\max} Z_{Ec}^{t,l}, \forall l \leq k \\
0 \leq P_{Ed}^{t,l} \leq P_{Ed}^{\max} Z_{Ed}^{t,l}, \forall l \leq k \\
Z_{Ec}^{t,l} + Z_{Ed}^{t,l} \leq 1, \forall l \leq k \\
SOC^{1,l} - SOC^{init} - P_{Ec}^{1,l} \eta^{Ec} + \frac{P_{Ed}^{1,l}}{\eta^{Ed}} = 0, \forall l \leq k \\
SOC^{t,l} - SOC^{t-1,l} - P_{Ec}^{t,l} \eta^{Ec} + \frac{P_{Ed}^{t,l}}{\eta^{Ed}} = 0, \forall l \leq k \\
SOC^{24,l} - SOC^{init} = 0, \forall l \leq k \\
SOC^{\min} \leq SOC^{t,l} \leq SOC^{\max}, \forall l \leq k \\
(6) - (8).
\end{cases}
\tag{23}
$$

where $k$ is the number of iterations. $\eta$ is an introduced auxiliary decision variable.

(2)  Subproblem of the Original Problem

After the decision being made in the first stage, the optimization results of the second stage can be acquired by solving the subproblem. At the $l$th iteration of the outer level, the optimal solution $\mathbf{y}_l$, $\mathbf{y}_l = [P_{Pb}^{t,l}, P_{Ps}^{t,l}, Z_{Pb}^{t,l}, Z_{Ps}^{t,l}]$, is already given. Then, the optimization results of the second stage can be acquired by solving the following subproblem:

$$
\begin{cases}
\max\limits_{\mathbf{U_{res}}, \mathbf{U_{Load}}} \min\limits_{\mathbf{X_{Non}^t}} \sum\limits_{t=1}^{T} c_E \left( P_{Ec}^t + P_{Ed}^t \right) \\
s.t.\ P_{Pb}^{t,l} + P_{res}^t + P_{Ed}^t - P_{Ec}^t - P_{Ps}^{t,l} - P_{Load}^t = 0 \\
(9) - (17).
\end{cases}
\tag{24}
$$

3.3.2. Inner-Level C&CG Procedures

Considering that the subproblem of the original problem is bi-level and the inner level is a non-convex optimization problem containing 0–1 decision variables, the subproblem is broken down into a subproblem and a main problem, which can be solved alternately to acquire the optimization results of the subproblem. The optimal solution $\mathbf{y}_l$ at the $l$th iteration in the outer-level C&CG process is already known. Then, the optimization results of the subproblem of the original problem at the $l$th iteration of the outer level can be obtained by the inner-level C&CG process.

(1)  Subproblem of the Subproblem

At the $r$th iteration of the inner level, the worst-case scenario of uncertain variables, denoted by $\mathbf{u}_r$, $\mathbf{u}_r = [P_{res}^{t,r}, P_{Load}^{t,r}]$, is given at the previous iteration. Then, the optimization results can be acquired by solving the following subproblem of the subproblem:

$$
\begin{cases}
\min\limits_{\mathbf{X_{Non}^t}} \sum\limits_{t=1}^{T} c_E \left( P_{Ec}^t + P_{Ed}^t \right) \\
s.t.\ P_{Pb}^{t,l} + P_{res}^{t,r} + P_{Ed}^t - P_{Ec}^t - P_{Ps}^{t,l} - P_{Load}^{t,r} = 0 \\
(9) - (15).
\end{cases}
\tag{25}
$$

(2)  Main Problem of the Subproblem

Since the feasible set of the 0–1 decision variable matrix $\mathbf{z}$ in the subproblem is bounded, $\mathbf{z}$ is denoted by $\{\mathbf{z}^1, \dots, \mathbf{z}^s\}$. At the $r$th iteration of the inner level, the optimal solution

$\mathbf{z}_r$, $\mathbf{z}_r = [Z_{Ec}^{t,r}, Z_{Ed}^{t,r}]$, is given by solving the subproblem of the subproblem. Then, decision variables $\mathbf{x}^r$, $\mathbf{x}^r = [P_{Ec}^{t,r}, P_{Ed}^{t,r}, SOC^{t,r}]$, and their corresponding constraints are added. Finally, the optimal solution $\mathbf{u}_r$ can be acquired by solving the following main problem of the subproblem:

$$
\begin{cases}
\max\limits_{\mathbf{U_{res}},\mathbf{U_{Load}}} \theta \\
s.t.\ \theta \leq \min\limits_{\mathbf{X_{Non}^t}} \sum\limits_{t=1}^{T} c_E\left(P_{Ec}^{t,r} + P_{Ed}^{t,r}\right), \forall r \leq m \\
P_{Pb}^{t,l} + P_{res}^t + P_{Ed}^{t,r} - P_{Ec}^{t,r} - P_{Ps}^{t,l} - P_{Load}^t = 0, \forall r \leq m \\
0 \leq P_{Ec}^{t,r} \leq P_{Ec}^{\max} Z_{Ec}^{t,r}, \forall r \leq m \\
0 \leq P_{Ed}^{t,r} \leq P_{Ed}^{\max} Z_{Ed}^{t,r}, \forall r \leq m \\
SOC^{1,r} - SOC^{init} - P_{Ec}^{1,r}\eta^{Ec} + \frac{P_{Ed}^{1,r}}{\eta^{Ed}} = 0, \forall r \leq m \\
SOC^{t,r} - SOC^{t-1,r} - P_{Ec}^{t,r}\eta^{Ec} + \frac{P_{Ed}^{t,r}}{\eta^{Ed}} = 0, \forall r \leq m \\
SOC^{24,r} - SOC^{init} = 0, \forall r \leq m \\
SOC^{\min} \leq SOC^{t,r} \leq SOC^{\max}, \forall r \leq m \\
(16) - (17),
\end{cases}
\tag{26}
$$

where $m$ is the number of iterations. $\theta$ is the introduced auxiliary decision variable.

It should be noted that there is a minimization problem in the constraints of the model (26), which is modeled as (A1) in Appendix A. Since the minimization problem (A1) is a linear programming problem, the Karush–Kuhn–Tucker (KKT) conditions can be leveraged to transform the model (26). The KKT conditions of the minimization problem (A1) are presented as (A2)–(A5) in Appendix A.

Since the complementary slackness conditions (A5) contain a large number of bilinear terms, the big-M method is used to linearize them. After linearization, the complementary slackness conditions (A5) are transformed as (A6) in Appendix B. The dual feasible conditions (A4) are also contained in (A6). Therefore, the dual feasible conditions (A4) and the complementary slackness conditions (A5) can be equivalently replaced by (A6) in Appendix B.

Finally, the model (26) can be eventually converted into a linear optimization problem which is presented as follows:

$$
\begin{cases}
\max\limits_{\mathbf{U_{res}},\mathbf{U_{Load}}} \theta \\
s.t.\ \theta \leq \sum\limits_{t=1}^{T} c_E\left(P_{Ec}^{t,r} + P_{Ed}^{t,r}\right), \forall r \leq m \\
(A2) - (A3), (B1) \\
(16) - (17).
\end{cases}
\tag{27}
$$

The main problem of the subproblem is readily solved by any available solver, for example, the Cplex solver of Matlab R2019b software.

### 3.4. The Iteration Procedure of the Introduced Nested C&CG Algorithm

To facilitate the application of the introduced algorithm, its iterative procedures are shown in Figure 3.

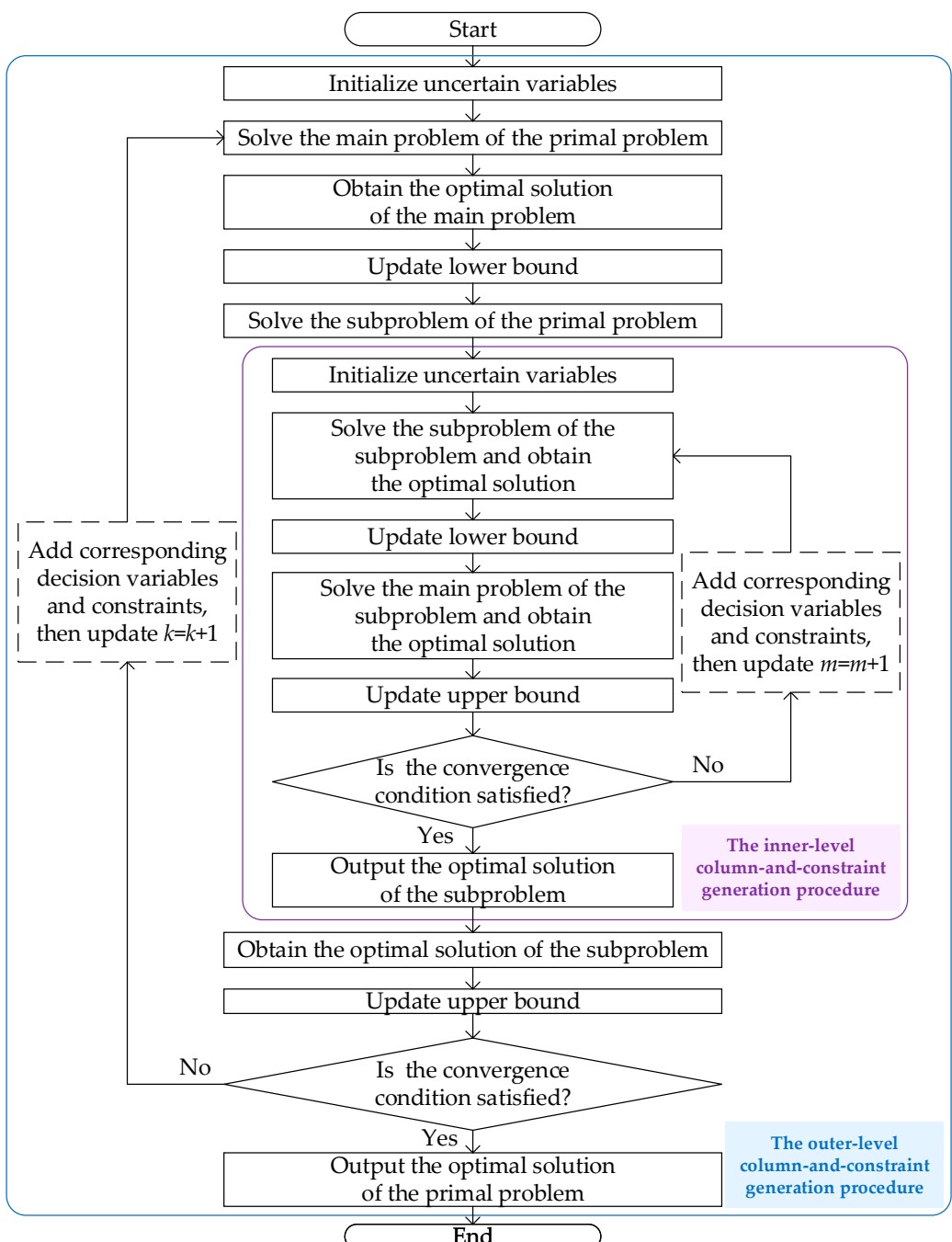

**Figure 3.** The iterative procedures of the introduced nested C&CG algorithm.

## 4. Case Study

In this paper, we select prosumers containing wind power, an ESS, and fixed load to verify the feasibility and effectiveness of the constructed model and algorithm. The parameters are shown in Table 1. The forecasted data and fluctuation range of wind power generation/load demand are presented in Figures 4 and 5. The purchase and sale prices of electricity in the electricity market are shown in Figure 6.

**Table 1.** Parameter setting.

| Parameters | Value of Parameters |
|---|---|
| Upper limit of purchased/sold power (kW) | 450/450 |
| Coefficient of charged and discharged cost of the ESS (CNY/kW) | 0.05 |
| Upper limit of charged/discharged power of the ESS (kW) | 100/100 |
| Initial quantity of electricity stored in the ESS (kWh) | 125 |
| Upper/lower limit of the capacity of the ESS (kWh) | 225/25 |
| Charged/discharged efficiency of the ESS | 0.98/0.98 |

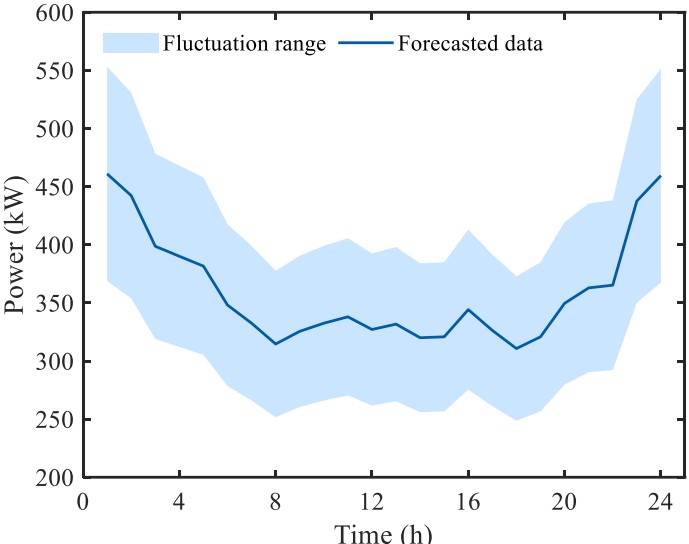

**Figure 4.** Forecasted data and fluctuation range of wind power generation.

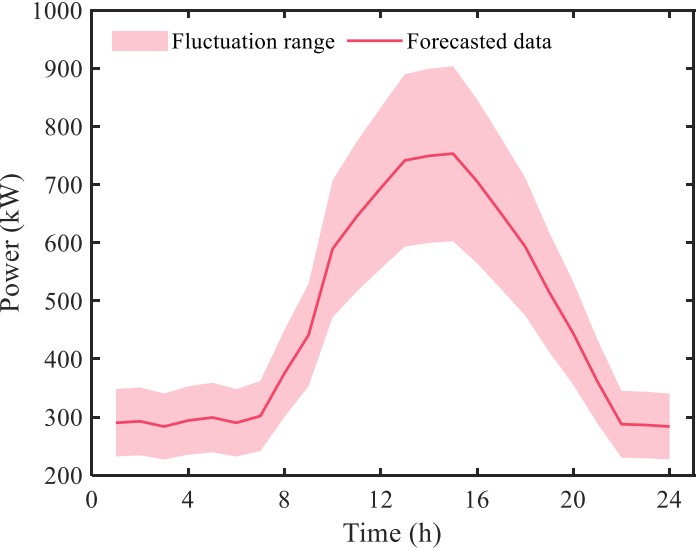

**Figure 5.** Forecasted data and fluctuation range of load demand.

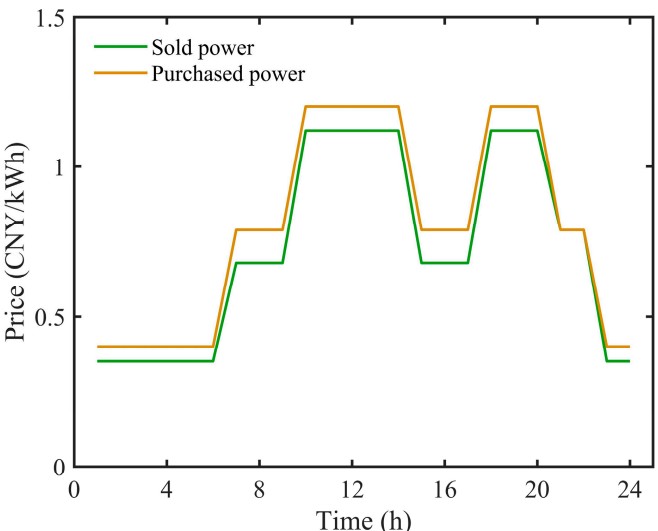

**Figure 6.** Purchase and sale prices of power in the electricity market.

### 4.1. Optimal Scheduling Results of Prosumers

In the case study, the robust parameter $\Gamma$ is set to 12. The final overall operating cost of prosumers is CNY 3201.03. The worst-case scenarios of multiple uncertainties caused by the wind power generation and the load demand are shown in Figures 7 and 8. The market trading strategy of prosumers is shown in Figure 9, where the positive/negative value represents the electricity purchased/sold by prosumers from/to the electricity market. The optimal scheduling results of the ESS is shown in Figure 10, where the positive/negative value represents the discharged/charged power of the ESS.

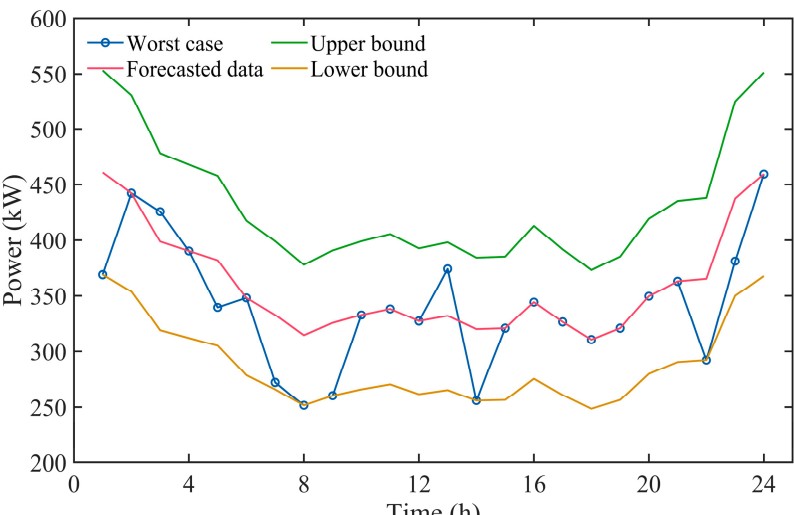

**Figure 7.** Worst-case scenario of wind power generation.

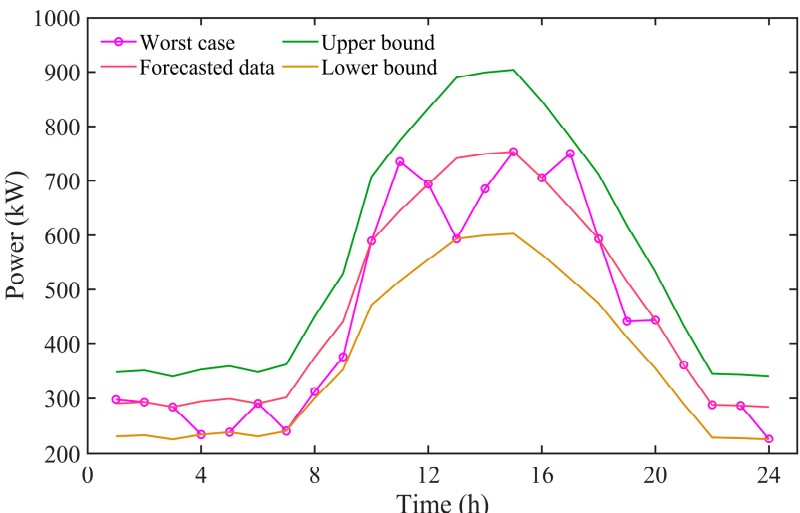

**Figure 8.** Worst-case scenario of load demand.

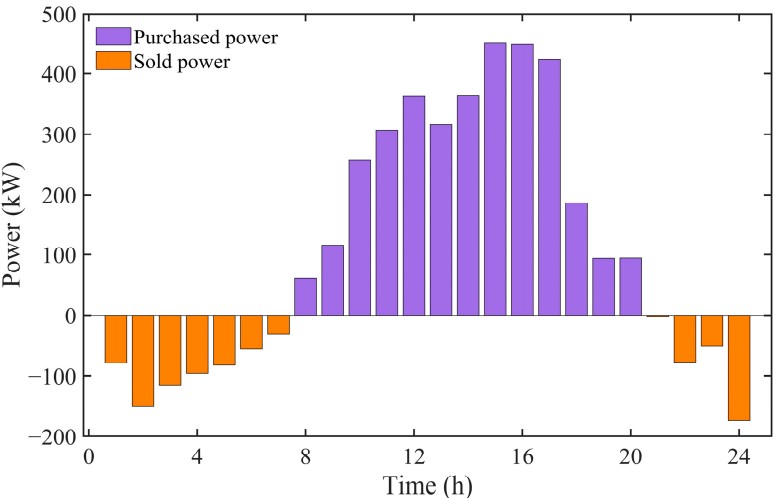

**Figure 9.** Market trading strategy.

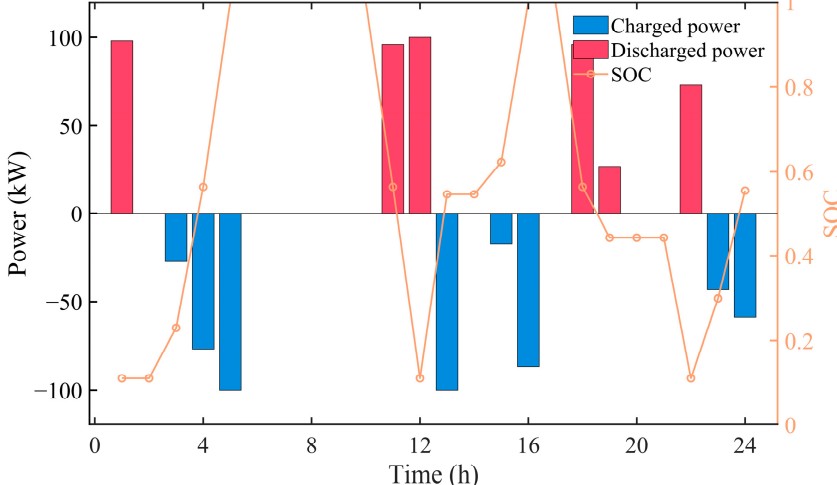

**Figure 10.** Optimal scheduling results of the ESS.

As seen from Figures 7–10, during the nighttime period (23:00–5:00 of the next day), the load demand of prosumers is at its valley period, and the wind power generation is at its peak period. The electricity price is comparatively low at this time period, so prosumers respond to the electricity price to charge the ESS on the whole, which can contribute to promoting the local accommodation of wind power during the nighttime period. Prosumers would sell their surplus electricity in the electricity market. During the daytime period (10:00–19:00), the load demand of prosumers is at its peak period, and the wind power generation is at its valley period. The electricity price is comparatively high at this time, so prosumers respond to the electricity price to discharge the ESS on the whole, which can compensate for the lack of wind power generation during the daytime period. Prosumers would purchase their insufficient electricity in the electricity market.

Prosumers respond to the electricity price to reasonably dispatch the ESS, charging the ESS in the valley period of the electricity price and discharging the ESS in the peak period of the electricity price. As a result, prosumers can utilize the peak–valley difference in the electricity price to arbitrage, thus reducing the overall operating cost. In addition, prosumers dispatch the ESS to store surplus wind power generation during the nighttime period and release the stored electricity when the wind power generation is insufficient during the daytime period. This can contribute to promoting the local accommodation of renewable energy and improving the efficiency of renewable energy utilization.

### 4.2. Optimal Scheduling Results of Prosumers under the Previous TSRO Model

To demonstrate the reasonability of the proposed TSRO model, the previous TSRO model of which the second-stage optimization is a convex optimization problem not containing 0–1 decision variables is solved using the C&CG algorithm. In the previous TSRO model, the ESS can be charged and discharged at the same time in the second-stage optimization. The robust parameter $\Gamma$ is also set to 12 for a fair comparison. The finally optimized overall operating cost of prosumers is CNY 1436.12. The market trading strategy of prosumers is shown in Figure 11, where the positive/negative value represents the electricity purchased/sold by prosumers from/to the electricity market. The optimal scheduling results of the ESS are shown in Figure 12, where the positive/negative value represents the discharged/charged power of the ESS.

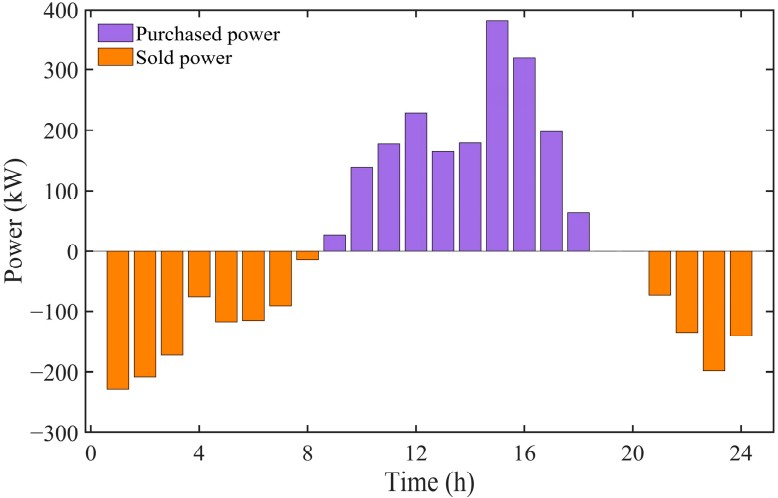

**Figure 11.** Market trading strategy under the previous TSRO model.

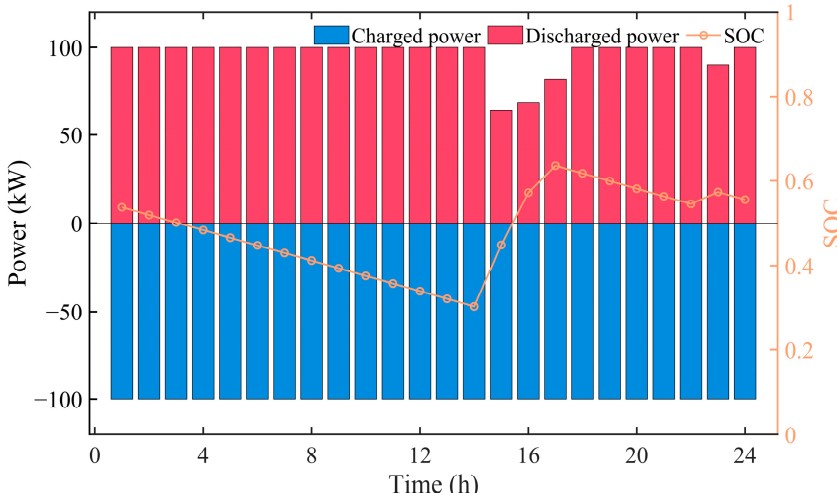

**Figure 12.** Optimal scheduling results of the ESS under the previous TSRO model.

As can be seen from the comparison between Figures 10 and 12, ESS cannot be charged and discharged simultaneously under the proposed model, while it can be charged and discharged simultaneously under the previous TSRO model. Moreover, as can be seen from the comparison between Figures 9 and 11, under the previous TSRO model, prosumers purchase less electricity from the electricity market and sell more electricity to the electricity market compared to those under the proposed TSRO model.

Compared to the proposed TSRO model, the overall operating cost of prosumers under the previous TSRO model has decreased from CNY 3201.03 to CNY 1436.12. This is because the previous TSRO model does not consider the case that the ESS cannot be charged and discharged simultaneously. Therefore, the overall operating cost of prosumers obtained from solving the previous TSRO model is smaller than that from solving the proposed TSRO model. However, it is unreasonable that the ESS can be charged and discharged simultaneously. Consequently, the proposed TSRO model is more reasonable than the previous TSRO model.

### 4.3. Influence of the Robust Parameter on the Market Trading Strategy and the Scheduling Results of the ESS

To analyze further the impact of multiple uncertainties caused by wind power generation and load demand on the market trading strategy and the scheduling results of the ESS, the robust parameter $\Gamma$ is set to 0, 6, and 12.

The market trading strategy of prosumers under different robust parameters is shown in Figure 13, where the positive/negative value represents the electricity purchased/sold by prosumers from/to the electricity market. It can be seen that the robust parameter has an impact on the market trading strategy of prosumers. As the robust parameter increases, prosumers purchase more electricity at 12:00–14:00 and sell more electricity at 5:00–6:00. This is because as the robust parameter increases, the uncertainties gradually increase, and prosumers need to purchase or sell more electricity to meet the internal energy supply and demand balance.

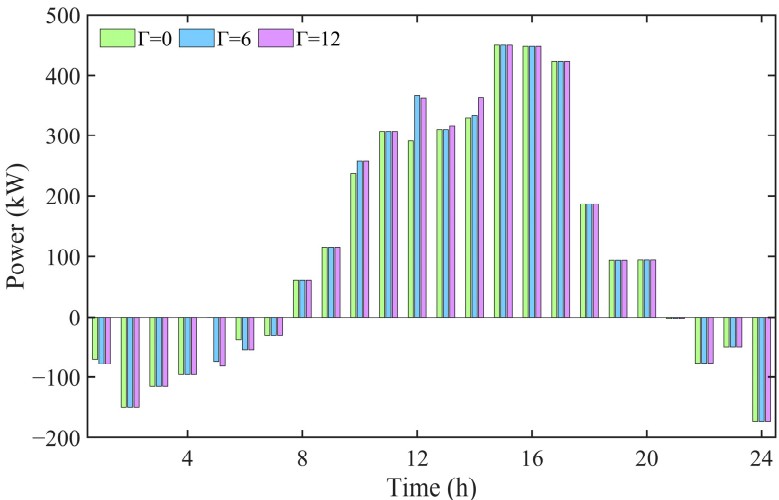

**Figure 13.** Market trading strategy under different robust parameters.

The optimal scheduling results of the ESS under different robust parameters is shown in Figure 14, where the positive/negative value represents the discharged/charged power of the ESS. It can be seen that the robust parameter has an impact on the optimal scheduling results of the ESS of prosumers. As the robust parameter increases, prosumers charge and discharge more frequently with increased fluctuation. This is because as the robust parameter increases, the uncertainties gradually increase; coupled with the lower charged/discharged cost of the ESS, prosumers invoke the ESS for charging and discharging more frequently to meet the internal energy supply and demand balance.

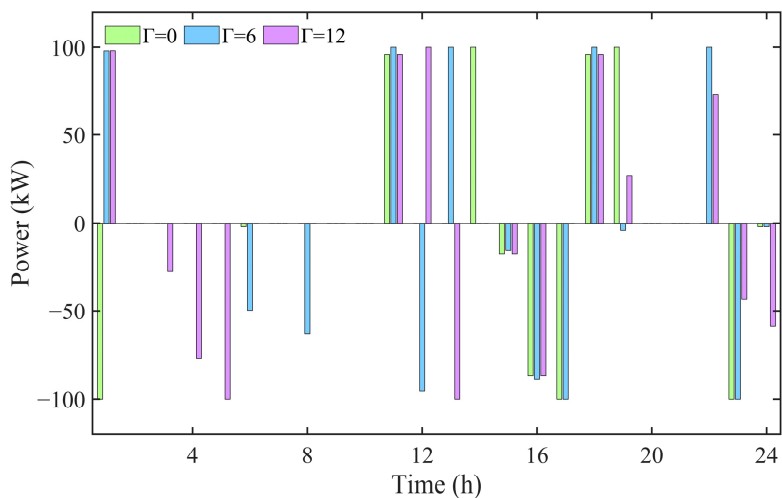

**Figure 14.** Optimal scheduling results of the ESS under different robust parameters.

Multiple uncertainties can increase the fluctuation of the market trading strategy and the scheduling results of the ESS, so it is necessary to use the robust optimization method to address the multiple uncertainties in the constructed model.

### 4.4. Performance Analysis of the Nested C&CG Algorithm

The robust parameter $\Gamma$ is set to 12. The final overall operating cost of prosumers is CNY 3201.03. The proposed TSRO model is run on a PC with a 12th Gen Intel Core i5-12500H CPU @ 2.50 GHz CPU and 16.0 GB RAM. The iteration process of the algorithm is shown in Table 2. As shown in Table 2, the outer-level procedure converges in only two iterations, and the inner-level procedure converges in only two iterations as well. So, the introduced algorithm requires only a small number of iterations to converge. When the

convergence accuracy is $10^{-6}$, the final solution time of the algorithm is only 9.75 s, which can fully satisfy the requirement of day-ahead scheduling for prosumers. It can be seen that the introduced algorithm which is simple to implement has high convergence accuracy and fast convergence speed.

It should be noted that although the existing C&CG algorithm can efficiently solve the existing TSRO model of which the second-stage optimization is a convex optimization problem not containing 0–1 decision variables, it cannot be applied directly to solve the proposed TSRO model of which the second-stage optimization is a bi-level one and the inner level is a non-convex optimization problem containing 0–1 decision variables.

**Table 2.** Iterative process of the nested C&CG algorithm.

| Outer Level | | | Inner Level | | |
|---|---|---|---|---|---|
| Iteration Ordinal Number | Lower Bound | Upper Bound | Iteration Ordinal Number | Lower Bound | Upper Bound |
| 1 | 3200.62 | 3201.03 | 1 | 40.0082 | 40.4164 |
| | | | 2 | 40.4164 | 40.4164 |
| 2 | 3201.03 | 3201.03 | 1 | 40.4164 | 50.0102 |
| | | | 2 | 50.0102 | 50.0102 |

*4.5. Discussion*

As can be seen from the aforementioned optimal scheduling results of prosumers, through reasonable dispatch of the ESS, prosumers can fully utilize the peak–valley difference in the electricity price to reduce the overall operating cost, as well as promote the local accommodation of renewable energy and improve the efficiency of renewable energy utilization.

In addition, by comparison with the optimal scheduling results of prosumers obtained from solving the previous TSRO model, it is known that the proposed TSRO model has considered the case where the ESS cannot be charged and discharged simultaneously. Therefore, the proposed TSRO model is more reasonable than the previous TSRO model, although the overall operating cost is relatively higher.

In addition, through comparison of the optimization results for prosumers under different robust parameters, it can be seen that multiple uncertainties can increase the fluctuation of the market trading strategy and the scheduling results of the ESS. Therefore, it is necessary to use the robust optimization method to address the multiple uncertainties in the proposed TSRO model.

Moreover, the introduced algorithm requires only four iterations to converge. The final solution time of the algorithm is only 9.75 s under the convergence accuracy of $10^{-6}$. The results indicate that the introduced algorithm has high convergence accuracy and fast convergence speed.

**5. Conclusions**

This paper constructs a TSRO model for prosumers under multiple uncertainties. In addition, considering that the second stage of the model is bi-level and the inner level is a non-convex optimization problem containing 0–1 decision variables, a nested C&CG algorithm is extended and used. Finally, a case study is solved. The results show that prosumers can dispatch the ESS reasonably to reduce the overall operating cost and promote the local accommodation of renewable energy. Compared to the previous TSRO model, the overall operating cost of prosumers under the proposed TSRO model has increased from CNY 1436.12 to CNY 3201.03. Although the overall operating cost under the proposed TSRO model is relatively higher, the proposed TSRO model has considered the scenarios where the ESS cannot be charged and discharged simultaneously, as can be seen from the case study result. Therefore, the proposed TSRO model is more reasonable than the previous TSRO model. In addition, the case study results show that the multiple

uncertainties have an impact on the market trading strategy and the optimal scheduling results of the ESS. Consequently, it is necessary to adopt the robust optimization method to address the uncertainties in the corresponding model. Moreover, the extended algorithm is simple to implement and requires only 4 iterations and a short computation time of 9.75 s to converge under the convergence accuracy of $10^{-6}$, enabling an efficient solution of the model.

However, the studies on the trading strategies of prosumers are limited to single electricity transactions between prosumers and the electricity market in this paper, without making full use of the complementary characteristics of multiple prosumers. In future work, electricity transactions between prosumers and the electricity market, as well as electricity transactions among multiple prosumers, could be considered in the optimization problem to fully utilize the complementary characteristics of the electricity consumption behavior of multiple prosumers.

**Author Contributions:** Conceptualization, P.G.; methodology, Q.Z. and J.Z.; validation, R.Z.; formal analysis, L.L.; investigation, S.W.; software, L.C. and W.W.; funding acquisition, S.Y. All authors have read and agreed to the published version of the manuscript.

**Funding:** This research was funded by State Grid Gansu Electric Power Company under grant 52272222000N.

**Institutional Review Board Statement:** Not applicable.

**Informed Consent Statement:** Not applicable.

**Data Availability Statement:** The data can be requested from the corresponding author by email.

**Conflicts of Interest:** The authors declare no conflict of interest.

## Nomenclature

| | |
|---|---|
| ESS | Energy storage system |
| PV | Photovoltaic |
| P2P | Peer-to-peer |
| RO/SO/DRO | Robust/stochastic/distributed robust optimization |
| TSRO | Two-stage robust optimization |
| C&CG | Column-and-constraint generation |
| $T$ | Index of hours |
| $k/m$ | Index of outer-level/inner-level iterations |
| $T$ | Set of time intervals |
| $\mathbf{U_{res}}/\mathbf{U_{Load}}$ | Set of wind power generation/load demand uncertainty |
| *Parameters* | |
| $\Gamma$ | Robust parameter |
| $\mu_{Ps}^t/\mu_{Pb}^t$ | Price for selling/purchasing electricity of prosumers at time $t$ |
| $c_E$ | Coefficient of the charged and discharged costs of the ESS |
| $P_{res,pre}^t/P_{Load,pre}^t$ | Forecasted data of wind power generation/load demand at time $t$ |
| $P_{Pb}^{\max}/P_{Ps}^{\max}$ | Maximum purchased/sold power of prosumers |
| $P_{Ec}^{\max}/P_{Ed}^{\max}$ | Maximum charged/discharged power of the ESS |
| $SOC^{init}$ | Initial quantity of electricity stored in the ESS |
| $SOC^{\max}/SOC^{\min}$ | Maximum/minimum value of the ESS capacity |
| $\eta^{Ec}/\eta^{Ed}$ | Charged/discharged efficiency of the ESS |
| $P_{Ps}^t/P_{Pb}^t$ | Electrical energy sold/purchased by prosumers at time $t$ |
| $P_{Ec}^t/P_{Ed}^t$ | Charged/discharged power of the ESS at time $t$ |
| $P_{res}^t/P_{Load}^t$ | Wind power generation/load demand of prosumers at time $t$ |
| $SOC^t$ | Quantity of electricity stored in the ESS at time $t$ |
| $Z_{Pb}^t/Z_{Ps}^t$ | State of purchasing/selling electricity for prosumers at time $t$ |
| $Z_{Ec}^t/Z_{Ed}^t$ | State of being charged/discharged of the ESS at time $t$ |

**Appendix A**

The minimization problem in the constraints of the model (26) is modeled as follows:

$$
\begin{cases}
\min_{\mathbf{X}_{\text{Non}}^{t}} \sum_{t=1}^{T} c_E \left( P_{Ec}^{t,r} + P_{Ed}^{t,r} \right), \forall r \leq m \\
s.t. \; P_{Pb}^{t,l} + P_{res}^{t} + P_{Ed}^{t,r} - P_{Ec}^{t,r} - P_{Ps}^{t,l} - P_{Load}^{t} = 0, \forall r \leq m \\
0 \leq P_{Ec}^{t,r} \leq P_{Ec}^{\max} Z_{Ec}^{t,r}, \forall r \leq m \\
0 \leq P_{Ed}^{t,r} \leq P_{Ed}^{\max} Z_{Ed}^{t,r}, \forall r \leq m \\
SOC^{1,r} - SOC^{init} - P_{Ec}^{1,r} \eta^{Ec} + \frac{P_{Ed}^{1,r}}{\eta^{Ed}} = 0, \forall r \leq m \\
SOC^{t,r} - SOC^{t-1,r} - P_{Ec}^{t,r} \eta^{Ec} + \frac{P_{Ed}^{t,r}}{\eta^{Ed}} = 0, \forall r \leq m \\
SOC^{24,r} - SOC^{init} = 0, \forall r \leq m \\
SOC^{\min} \leq SOC^{t,r} \leq SOC^{\max}, \forall r \leq m.
\end{cases}
\tag{A1}
$$

The standard KKT conditions consist of the primal feasible conditions, the first-order stationary conditions, the dual feasible conditions, and the complementary slackness conditions. The KKT conditions of the minimization problem (A1) are presented as follows:

(1)    The primal feasible conditions

$$
\begin{cases}
P_{Pb}^{t,l} + P_{res}^{t} + P_{Ed}^{t,r} - P_{Ec}^{t,r} - P_{Ps}^{t,l} - P_{Load}^{t} = 0, \forall r \leq m \\
0 \leq P_{Ec}^{t,r} \leq P_{Ec}^{\max} Z_{Ec}^{t,r}, \forall r \leq m \\
0 \leq P_{Ed}^{t,r} \leq P_{Ed}^{\max} Z_{Ed}^{t,r}, \forall r \leq m \\
SOC^{1,r} - SOC^{init} - P_{Ec}^{1,r} \eta^{Ec} + \frac{P_{Ed}^{1,r}}{\eta^{Ed}} = 0, \forall r \leq m \\
SOC^{t,r} - SOC^{t-1,r} - P_{Ec}^{t,r} \eta^{Ec} + \frac{P_{Ed}^{t,r}}{\eta^{Ed}} = 0, \forall r \leq m \\
SOC^{24,r} - SOC^{init} = 0, \forall r \leq m \\
SOC^{\min} \leq SOC^{t,r} \leq SOC^{\max}, \forall r \leq m.
\end{cases}
\tag{A2}
$$

(2)    The first-order stationary conditions

$$
\begin{cases}
c_E - \mu_e^{t,r} - \eta^{Ec} \mu_Q^{t,r} + \lambda_{Ec1}^{t,r} - \lambda_{Ec2}^{t,r} = 0, \forall r \leq m \\
c_E + \mu_e^{t,r} + \frac{\mu_Q^{t,r}}{\eta^{Ed}} + \lambda_{Ed1}^{t,r} - \lambda_{Ed2}^{t,r} = 0, \forall r \leq m \\
\mu_Q^{t,r} - \mu_Q^{t+1,r} + \lambda_{Q1}^{t,r} - \lambda_{Q2}^{t,r} = 0, \forall r \leq m \\
\mu_Q^{24,r} + \mu_Q^{25,r} + \lambda_{Q1}^{24,r} - \lambda_{Q2}^{24,r} = 0, \forall r \leq m,
\end{cases}
\tag{A3}
$$

where $\lambda_{Ec1}^{t,r}$, $\lambda_{Ec2}^{t,r}$, $\lambda_{Ed1}^{t,r}$, $\lambda_{Ed2}^{t,r}$, $\lambda_{Q1}^{t,r}$, $\lambda_{Q2}^{t,r}$, $\mu_e^{t,r}$, and $\mu_Q^{t,r}$ are dual variables.

(3)    The dual feasible conditions

$$
\left\{ \lambda_{Ec1}^{t,r}, \lambda_{Ec2}^{t,r}, \lambda_{Ed1}^{t,r}, \lambda_{Ed2}^{t,r}, \lambda_{Q1}^{t,r}, \lambda_{Q2}^{t,r} \geq 0, \forall r \leq m \right.
\tag{A4}
$$

(4)    The complementary slackness conditions

$$
\begin{cases}
\lambda_{Ec1}^{t,r} \left( P_{Ec}^{t,r} - P_{Ec}^{\max} Z_{Ec}^{t,r} \right) = 0, \forall r \leq m \\
\lambda_{Ec2}^{t,r} \left( -P_{Ec}^{t,r} \right) = 0, \forall r \leq m \\
\lambda_{Ed1}^{t,r} \left( P_{Ed}^{t,r} - P_{Ed}^{\max} Z_{Ed}^{t,r} \right) = 0, \forall r \leq m \\
\lambda_{Ed2}^{t,r} \left( -P_{Ed}^{t,r} \right) = 0, \forall r \leq m \\
\lambda_{Q1}^{t,r} \left( SOC^{t,r} - SOC^{\max} \right) = 0, \forall r \leq m \\
\lambda_{Q2}^{t,r} \left( -SOC^{t,r} + SOC^{\min} \right) = 0, \forall r \leq m.
\end{cases}
\tag{A5}
$$

**Appendix B**

After linearization with the big-M method, the complementary slackness conditions (A5) are transformed as follows:

$$
\begin{cases}
\lambda_{Ec1}^{t,r} \le MZ_{Ec1}^{t,r}, \forall r \le m \\
-M\left(1 - Z_{Ec1}^{t,r}\right) \le P_{Ec}^{t,r} - P_{Ec}^{\max} Z_{Ec}^{t,r}, \forall r \le m \\
\lambda_{Ec2}^{t,r} \le MZ_{Ec2}^{t,r}, \forall r \le m \\
P_{Ec}^{t,r} \le M\left(1 - Z_{Ec2}^{t,r}\right), \forall r \le m \\
\lambda_{Ed1}^{t,r} \le MZ_{Ed1}^{t,r}, \forall r \le m \\
-M\left(1 - Z_{Ed1}^{t,r}\right) \le P_{Ed}^{t,r} - P_{Ed}^{\max} Z_{Ed}^{t,r}, \forall r \le m \\
\lambda_{Ed2}^{t,r} \le MZ_{Ed2}^{t,r}, \forall r \le m \\
P_{Ed}^{t,r} \le M\left(1 - Z_{Ed2}^{t,r}\right), \forall r \le m \\
\lambda_{Q1}^{t,r} \le MZ_{Q1}^{t,r}, \forall r \le m \\
-M\left(1 - Z_{Q1}^{t,r}\right) \le SOC^{t,r} - SOC^{\max}, \forall r \le m \\
\lambda_{Q2}^{t,r} \le MZ_{Q2}^{t,r}, \forall r \le m \\
SOC^{t,r} - SOC^{\min} \le M\left(1 - Z_{Q2}^{t,r}\right), \forall r \le m,
\end{cases}
\tag{A6}
$$

where $M$ is a sufficiently large constant. $Z_{Ec1}^{t,r}$, $Z_{Ec2}^{t,r}$, $Z_{Ed1}^{t,r}$, $Z_{Ed2}^{t,r}$, $Z_{Q1}^{t,r}$, and $Z_{Q2}^{t,r}$ are the introduced auxiliary 0–1 decision variables.

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
