# Peer review of "Two-Stage Robust Optimization for Prosumers Considering Uncertainties from Sustainable Energy of Wind Power Generation and Load Demand Based on Nested C&CG Algorithm"

_sustainability, doi:10.3390/su15129769_

Round 1

Reviewer 1 Report

This paper develops the optimization model to solve the uncertainty situation. It is well written and organized.  The results are clearly presented. There are a few comments and suggestions remaining.

1. The contributions/novelty of the article has not been well highlighted in the manuscript. It should be highlighted at the end of the introduction section. 

2. Was it not possible to achieve the goals with simpler controllers?

3. Introduction must be improved. It is not enough to state the current work and should be expanded. More related publications should be included in the literature review to improve the state-of-the-art in the field. Here are suggested papers: 

https://doi.org/10.1016/j.rser.2018.03.047
https://doi.org/10.3390/en12244817

https://doi.org/10.1049/iet-rpg.2016.0277   4. The motivation of the research is not clear and the innovation of the paper is insufficient, if it is not then these should be respectively given.

5. What about the performance of the proposed algorithm compared with others existing in the literature? The recommended method should be presented in comparison with many other publications in the literature.

Just minor errors

Reviewer 2 Report

In this paper, the authors presented a two-stage robust optimization model for prosumers under multiple uncertainties. In addition, the nested C&CG algorithm has been introduced considering that the second stage of the model is bi-level and the inner level is a non-convex optimization problem containing 0-1 decision variables. The paper is organized and written well, and the topic is interesting and deserves attention. However, there are many concerns that must be considered:

1)      The abstract is very general; numbers or percentages about the main results and the main conclusion of the results must be included in the abstract.

2)      It is recommended to add a nomenclature table before the Introduction section to define all symbols and abbreviations.

3)      Please, insert the reference citation before all equations in the text.

4)      The paper organization should be added at the end of the introduction section.

5)      I would like to see the final value of the objective function comparing the proposed optimization model with any other previous model, if possible.

6)      The reviewer doubts whether the problem is still solvable by the commercial solver when the problem size becomes larger. In addition, the authors should also indicate how much time is needed to solve the problems presented in the case study.

7)      More recent references, which are mainly related to the study and considered multiple uncertainties in their models, are recommended to be added in the Introduction section like; An optimal network constraint-based joint expansion planning model for modern distribution networks with multi-types intermittent RERs; A multiple uncertainty-based Bi-level expansion planning paradigm for distribution networks complying with energy storage system functionalities; Joint expansion planning of distribution network with uncertainty of demand load and renewable energy; A novel multi-objective scheduling model for grid-connected hydro-wind-PV-battery complementary system under extreme weather: A case study of Sichuan, China; A novel optimization strategy for line loss reduction in distribution networks with large penetration of distributed generation.

8)      A separate discussion section is required before the Conclusions section to comment on the produced results.

9)      The conclusion section should be improved by quantifying the best results. Also, the authors are required to give suggestions for future work and limitations of the current study in the conclusions section.

Minor editing of English language required.

Reviewer 3 Report

The paper is well written and organized. This reviewer has the following comments:

1)      Can you add a Section 2 where you discuss about your proposed methodology. This section should be made with simple block diagrams and provide a good overview of your proposed method to the readers.

2)      Please provide the computer specifications you used to solve the proposed method.

Round 2

Reviewer 1 Report

The manuscript is significantly improved, and can be accepted for publication.

Reviewer 2 Report

The authors satisfactorily considered my comments. Only the authors should add references before the equations which have been mentioned before in the literature.